# Visualizing symmetry-breaking electronic orders in epitaxial Kagome magnet FeSn films

Huimin Zhang[1,2], Basu Dev Oli[1], Qiang Zou[1], Xu Guo[3], Zhengfei Wang [3] & Lian Li [1] ✉

Kagome lattice hosts a plethora of quantum states arising from the interplay of topology, spin-orbit coupling, and electron correlations. Here, we report symmetry-breaking electronic orders tunable by an applied magnetic field in a model Kagome magnet FeSn consisting of alternating stacks of two-dimensional $Fe_3Sn$ Kagome and $Sn_2$ honeycomb layers. On the $Fe_3Sn$ layer terminated FeSn thin films epitaxially grown on $SrTiO_3(111)$ substrates, we observe trimerization of the Kagome lattice using scanning tunneling micro-scopy/spectroscopy, breaking its six-fold rotational symmetry while preser-ving the translational symmetry. Such a trimerized Kagome lattice shows an energy-dependent contrast reversal in $dI/dV$ maps, which is significantly enhanced by bound states induced by Sn vacancy defects. This trimerized Kagome lattice also exhibits stripe modulations that are energy-dependent and tunable by an applied in-plane magnetic field, indicating symmetry-breaking nematicity from the entangled magnetic and charge degrees of freedom in antiferromagnet FeSn.

Kagome lattice, a two-dimensional hexagonal network of corner-sharing triangles (Fig. 1a), exhibits linearly dispersing Dirac cones at the Brillouin zone (BZ) corner K point and flat band (FB) through the rest of the BZ (Fig. 1b)[1]. These bands have been observed by angle-resolved photoemission spectroscopy (ARPES) in binary Kagome metal magnets $T_mX_n$ (T: 3$d$ transition metals, X: Sn, Ge, m:n = 3:1, 3:2, 1:1)[2,3], and ternary ferromagnetic $YMn_6Sn_6$[4]. Evidence of flat bands has been reported in FeSn thin films grown on $SrTiO_3(111)$ (STO) substrates in three-terminal planar Schottky tunneling measurements[5]. The interplay of spin-orbit-coupling and out-of-plane ferromagnetic order can further lead to Chern topological fermions, which have been observed in $TbMn_6Sn_6$[6]. The band structure of the Kagome lattice also exhibits saddle points at the BZ boundary M[1,4,7], which can lead to charge instabilities and symmetry-breaking electronic orders, includ-ing charge density waves (CDWs)[7], bond order waves (BOWs)[8–13], and

chiral superconductivity[14]. pair density waves[15] and (2 × 2) CDWs[16,17] have been reported in non-magnetic Kagome metals $AV_3Sb_5$ (A = K, Cs) and Kagome magnet FeGe[18–20]. The origins of the CDWs have been intensely debated. Rotational symmetry breaking driven by correlation and formation of chiral charge order have been suggested for the (2 × 2) CDWs observed on the Sb-terminated surface of the cleaved $KV_3Sb_5$ crystal[16,17]. On the Cs termination of the cleaved $CsV_3Sb_5$, an electronic nematicity driven by the (2 × 2) CDW is also reported[17].

Here, we report symmetry-breaking nematicity in a model Kagome magnet FeSn from the entangled magnetic and charge degrees of freedom. The crystal structure of FeSn (space group P6/mmm with lattice constant $a = 5.3$ Å, $c = 4.4$ Å[21]) is layered, consisting of alternatingly stacked planes of two-dimensional (2D) $Fe_3Sn$ Kagome (K layer) and $Sn_2$ honeycomb (S layer) (Fig. 1c inset). The material is also magnetically ordered, with Fe moments

[1]Department of Physics and Astronomy, West Virginia University, Morgantown, WV 26506, USA. [2]State Key Laboratory of Structural Analysis, Optimization and CAE Software for Industrial Equipment, Dalian University of Technology, Dalian 116024, China. [3]Hefei National Research Center for Physical Sciences at the Microscale, CAS Key Laboratory of Strongly-Coupled Quantum Matter Physics, Department of Physics, Hefei National Laboratory, University of Science and Technology of China, Hefei 230026 Anhui, China. ✉e-mail: lian.li@mail.wvu.edu

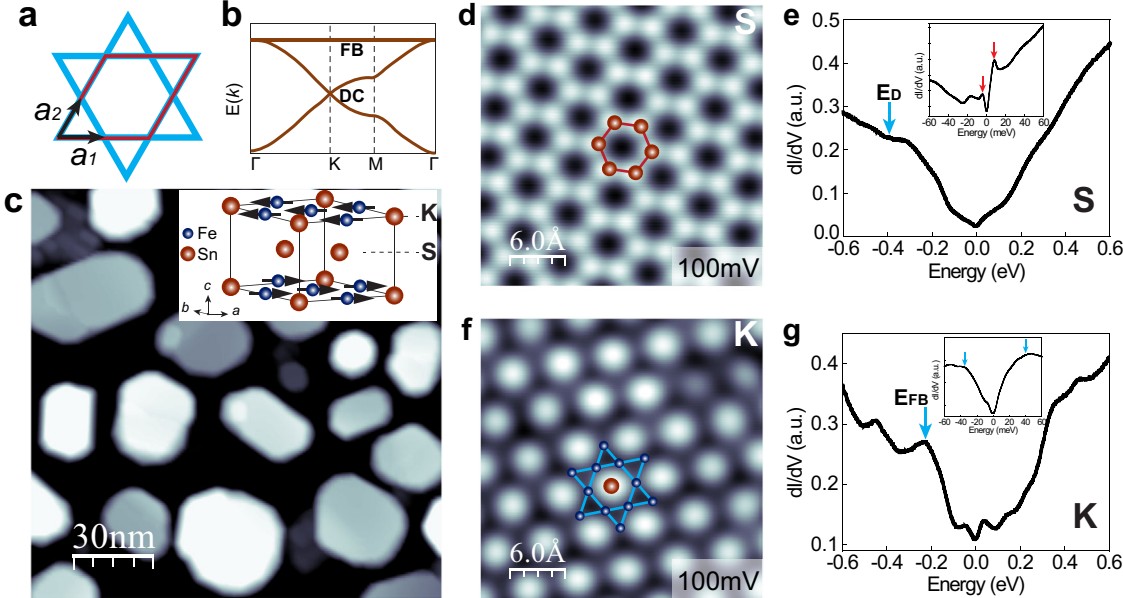

**Fig. 1 | FeSn films grown by MBE on the SrTiO₃(111) substrate with both Sn₂- and Fe₃Sn-terminations. a** Model of a 2D Kagome lattice unit cell with a triangular Bravais lattice with a 3-point basis $a_1$, $a_2$, and $a_3 = a_2 - a_1$. The unit cell contains three plaquettes (two triangles and one hexagon). **b** Schematic diagram of the tight binding band structure for the Kagome lattice, showing Dirac point at the K points, a flat band through the whole BZ, and saddle points at M. **c** Topographic STM image of epitaxial FeSn films. Setpoint: $V = 5.0$ V, $I = 5$ pA. Inset: schematic crystal structure of FeSn with the Fe₃Sn Kagome layer (K) and Sn honeycomb layer (S) marked.

**d** Atomic resolution STM image of the S layer exhibiting a honeycomb lattice. Setpoint: $V = 0.2$ V, $I = 3.0$ nA. **e** Typical dI/dV spectrum of the S layer. The cyan arrow marks the Dirac point $E_D = -0.36$ eV. Inset: close-up view of an asymmetric gap near $E_F$, bounded by two peaks with different intensities and energy positions at −4.0 and 7.8 meV (marked by red arrows). **f** Atomic resolution STM image of the K layer showing a close-packed lattice. Setpoint: $V = 0.1$ V, $I = 3.0$ nA. **g** Typical dI/dV spectrum of the K layer taken at the Sn site. The cyan arrow marks the flat band $E_{FB} = -0.23$ eV. Inset: a gap of ~80 meV is observed close to the Fermi level.

ferromagnetically aligned in the K layer and antiferromagnetically coupled between the K layers, with a Neel temperature $T_N = 368$ K[22]. The minimal coupling between K layers gives rise to a quasi-2D electronic structure, thus providing an ideal platform to probe the correlated states of Kagome lattice[2]. On the epitaxial FeSn films grown on STO(111) substrates by molecular beam epitaxy (MBE), we observe a honeycomb lattice on the Sn-termination and symmetry-breaking trimerization of the Kagome lattice on the Fe₃Sn layer using scanning tunneling microscopy/spectroscopy (STM/S). This trimerized structure shows an energy-dependent high/low contrast reversal of adjacent triangles of the Kagome lattice. Furthermore, the trimerized Kagome lattice also exhibits stripe modulations that are energy-dependent and tunable by an applied in-plane magnetic field, indicating symmetry-breaking nematicity from the interplay of magnetic and charge degrees of freedom in antiferromagnet FeSn.

## Results

### Epitaxial growth and electronic structure of FeSn

The films are grown on SrTiO₃(111) substrates that are thermally treated in ultrahigh vacuum to obtain step-terrace morphology with a (4 × 4) reconstruction (Supplementary Fig. S1). With a lattice mismatch of 4.0% between SrTiO₃(111) substrate ($a_{STO(111)} = 5.52$ Å) and FeSn ($a_{FeSn} = 5.30$ Å), three-dimensional island growth mode is observed at the initial stages. The topographic STM image in Fig. 1c reveals a surface morphology characterized by flattop islands with heights varying from 6 to 8 nm and lateral sizes typically below 50 nm. As the growth proceeds, smaller islands coalesce and form larger ones with lateral sizes of ~100 nm. X-ray diffraction measurements confirm the FeSn phase (Supplementary Fig. S2), similar to earlier reports of MBE-grown FeSn(001) thin films on SrTiO₃(111)[5,23] and LaAlO₃(111) substrates[24].

Atomic resolution STM imaging further reveals two terminations, similar to that produced by cleavage of bulk materials[2], indicative of the weak coupling between the Sn and Fe₃Sn layers in FeSn. The first exhibits a perfect honeycomb with a lattice constant $a = 0.53$ nm

(Fig. 1d), consistent with the Sn termination, as observed in CoSn[25]. However, it contrasts Fe₃Sn₂, where a buckled honeycomb structure was observed on the Sn-termination[3,26]. Furthermore, the dI/dV spectrum, which is proportional to the local density of states (LDOS), shows a dip at −0.36 eV (Fig. 1e), attributed to the Dirac point ($E_D$) based on comparison with ARPES measurements on the surface of cleaved bulk FeSn[2]. A small gap of 11.8 meV is also observed around the Fermi level (inset, Fig. 1e). The gap is bounded by two asymmetric peaks not only in intensity but also in energy positions, with the left peak at −4.0 meV and the right at 7.8 meV (red arrows). The nature of this gap is unknown, which could be attributed to electron interactions similar to that reported in 2D topological insulators 1T′-WTe₂[27].

For the second type of termination, a close-packed structure with a periodicity of 0.53 nm is observed (Fig. 1f), which is determined to be the Fe₃Sn Kagome layer. The distinct electronic properties of this structure are also reflected in the dI/dV spectrum, where a pronounced peak at −0.23 eV is observed (cyan arrow in Fig. 1g). This energy position is consistent with that of the flat band measured by ARPES on the Fe₃Sn termination of cleaved bulk FeSn[2], and planar tunneling measurements on epitaxial thin films of FeSn grown on STO(111)[5]. Near the Fermi level, a gap of ~80 meV is seen (marked by cyan arrows in Fig. 1g inset), which barely changes with different setpoints (Supplementary Fig. S3).

### STM imaging of the Sn₂-honeycomb and Fe₃Sn-Kagome layers

The Sn₂ honeycomb lattice shows minimum bias dependence between ±0.5 V (Supplementary Fig. S4). Figure 2a presents images taken near the Fermi level, where a uniform honeycomb is observed for all energies. This is in contrast to the surface of cleaved Fe₃Sn₂ bulk crystal, where the honeycomb structure of the Sn termination exhibits strong bias dependence[3,26], providing additional support for the FeSn film grown in this work. On the other hand, a significant bias dependence is apparent for the Fe₃Sn Kagome layer, particularly near the Fermi energy, as shown in Fig. 2b and Supplementary Fig. S5. While the close-packed structure is seen for bias between −100 and −10 mV and higher

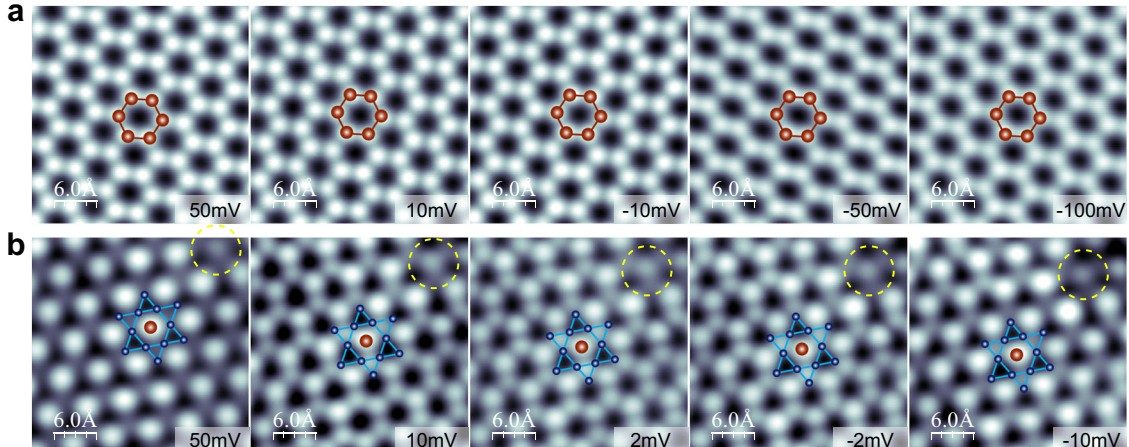

**Fig. 2 | Bias-dependent STM imaging of the Sn₂-honeycomb (S) and Fe₃Sn-Kagome (K) layers. a** Atomic resolution STM images of the Sn₂-honeycomb on the Sn₂- terminated FeSn at the bias voltages specified. Setpoint: $I$ = 5.0 nA. **b** Bias-dependent STM images of the Fe₃Sn-Kagome layer of Fe₃Sn-termination at the bias voltages specified. In both cases, ball-and-stick models of the honeycomb and the Kagome lattice are overlaid on top. Setpoint: $I$ = 3.0 nA. To account for the slight thermal draft from one image to the next, a Sn vacancy circled in (**b**) is used as a reference to assign the atomic lattice.

than 30 mV, a buckled honeycomb structure is observed near the Fermi level between 10 and −5 mV. Note that the bright contrast corresponding to the Sn atom at the center of the hexagon remains unchanged at all biases; while there is no observable difference at 100 mV, the up and down Fe triangles exhibit alternating high and low contrast at energies close to the Fermi level (Fig. 2b).

The assignment of the Sn (honeycomb) and Fe₃Sn (Kagome) termination is further confirmed by density-functional theory (DFT) calculations. Simulated STM images indicate that the Sn termination is a honeycomb structure within all energy ranges (Supplementary Fig. S6), consistent with experimental observations (Fig. 2a and Supplementary Fig. S4). In contrast, the Fe₃Sn termination shows strong energy dependence (Supplementary Fig. S7), further confirming the FeSn phase synthesized in this work. We note that there are recent studies on similar systems where the central Sn/Ge atom is often not visible and appears as low contrast in the STM images[16,20,28]. However, these prior studies are on the surface of cleaved bulk materials. Here, our samples are films grown by MBE under Sn-rich conditions (Sn/Fe flux ratio >3), likely leading to a different atomic registry on the Fe₃Sn terminated surface.

### Trimerization of the Fe₃Sn Kagome layer

We further carried out dI/dV mapping of the Fe₃Sn Kagome layer to examine the origin of the energy-dependent contrasts in the up and down triangles in STM imaging. First, typical dI/dV spectra were obtained at the up-triangle site (A), down-triangle site (B), and the central Sn site (C) (Fig. 3a, b). Interestingly, the dI/dV intensities for the up- and down-triangles cross at 16.8 meV (marked by a red arrow in Fig. 3b), below which A site has higher intensity. At ~100 meV, the intensity of the C site is the highest. These transitions are directly reflected in the energy-dependent dI/dV mapping (Fig. 3c–l and additional data in Supplementary Fig. S8). Compared to the topographic image (Fig. 3a), the dI/dV maps in Fig. 3c–l reveal trimerization of the Kagome lattice, where the up and down triangles exhibit different contrasts. Below the crossing energy of 16.8 meV, the up-triangles have a much higher contrast compared to the down-triangles. Above the transition, the contrast reverses with the down triangle exhibiting higher contrast. At ~100 meV, the central Sn site has the highest contrast, while the A and B sites are roughly similar. This energy-dependent contrast reversal between the up and down triangles indicates the trimerization of the Kagome lattice, breaking its six-fold rotational and mirror symmetry but not translational symmetry, as schematically illustrated in the inset of Fig. 3b.

### Enhanced trimerization near single Sn vacancy

Additional evidence for this symmetry-breaking state is found near Sn vacancy defects, the most common type of defects on the Fe₃Sn Kagome layer, as schematically shown in Fig. 4a and Supplementary Fig. S9. First, the topographic STM image indicates suppressed DOS at the Sn vacancy site (Fig. 4b), consistent with our DFT calculations (Supplementary Fig. S10). In the dI/dV map (Fig. 4c), the contrast of the up-triangle sites is enhanced significantly due to Sn-vacancy induced bound states, resulting in a three-lobe feature that can be more clearly seen by overlaying a Kagome lattice model on the image (Supplementary Fig. S11). This enhanced trimerization is also energy- and site-dependent. As shown in Fig. 4d–f, the dI/dV spectra taken on the three lobes reveal clear bound states at −49.8, −21.3, and −18.7 meV, respectively. The corresponding dI/dV maps recorded at these energies are shown in Fig. 4g–i and Supplementary Fig. S12. Three-fold symmetry is observed in dI/dV maps in the energy range [−200, −32 meV]. Interestingly, in maps with energy closer to the Fermi level, e.g., $g(\mathbf{r}, -21.3 \text{ meV})$ and $g(\mathbf{r}, -18.7 \text{ meV})$, the intensity at one of the up-triangles is suppressed, thus exhibiting a two-fold symmetry.

### Tuning the stripe modulations by an in-plane magnetic field

Given that Fe atoms in the Kagome layer are ordered ferromagnetically, we now examine the impact of the magnetic field on trimerization. Note that non-spin-polarized tips are used for STM imaging and spectroscopy; magnetic information is not expected directly. Nevertheless, as presented below, we have observed energy-dependent stripe modulations of the trimerized Kagome lattice.

Figure 5a shows an STM topography image and corresponding dI/dV maps taken at the energies specified without a magnetic field, where a close-packed structure is observed. While the close-packed structure is further confirmed by the FFT patterns shown in Fig. 5b, slight variations are present in the FFT peak intensities, indicative of a stripe modulation. This is not simply due to an asymmetrical tip, as confirmed by additional energy-dependent STM imaging (Supplementary Fig. S13).

With an applied 2 T in-plane magnetic field along the direction shown by the red arrow (Fig. 5c), the topography image is more symmetrical. However, the energy-dependent stripe modulations become more pronounced in the dI/dV maps. At −144 meV, the close-packed structure is modulated along one of the crystallographic directions, leading to substantially enhanced intensity at two of the peaks in the FFT pattern (second panel in Fig. 5d). Interestingly, a honeycomb structure appears at −98.7 meV, leading to a symmetrical FFT pattern.

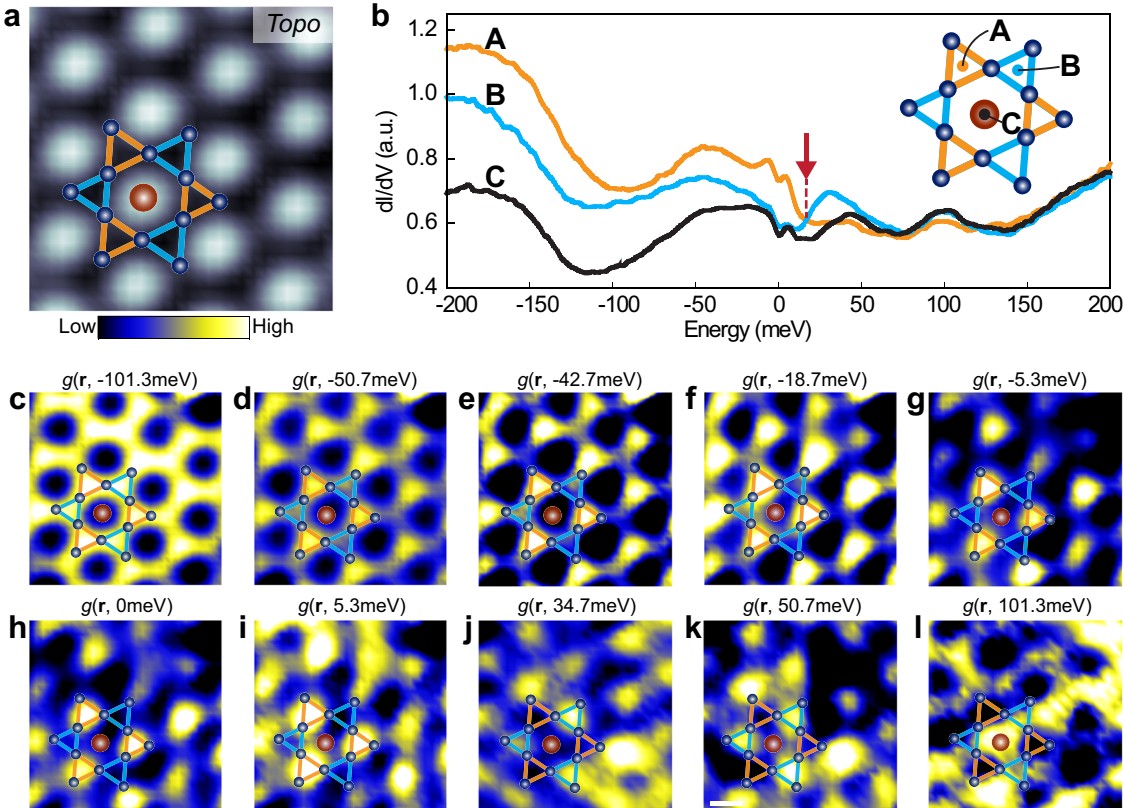

**Fig. 3 | Trimerized Kagome lattice on the Fe₃Sn layer. a** Topographic STM image of the Fe₃Sn layer. Setpoint: $V = 0.2$ V, $I = 5.0$ nA. **b** dI/dV spectra taken at three representative sites labeled A, B, and C (inset). The red arrow marks the intensity reversal for the up-and down-triangles at 16.8 meV. **c–l** Differential conductance maps of the same location at the energies marked. Setpoint: $V = 0.2$ V, $I = 5.0$ nA. A ball-and-stick model of the Kagome lattice is overlaid on the STM image (**a**) and differential conductance maps (**c–l**).

Then, the structure reverts to be close-packed at −50.7 meV (right panel in Fig. 5c), but with a stripe modulation rotated 60° from that at −144 meV. This stripe formation is again evident in the asymmetrical FFT patterns (Fig. 5d, right panel). Additional dI/dV maps in the energy window of −144 to 144 meV are provided in the Supplementary Information Figs. S14–16 to support this observation. While stripe modulations are also seen for a 9 T out-of-plane magnetic field, no clear energy dependence is detected.

To quantify the evolution of the stripe modulation, the energy-dependent FFT peak intensity is plotted in Fig. 5g and Supplementary Fig. S17. Along the direction marked by a white dotted line in Fig. 5d second panel, the maximum peak intensity shifts from ~50 meV below Fermi level at zero field to ~−100 meV at 2 T in-plane field. Interestingly, this shift is also remanent after removing the magnetic field. However, such behavior is not observed in the other two directions (Supplementary Fig. S17). These results clearly show a stripe modulation highly tunable by the magnetic field, indicating the coupling of magnetism with the charge order of the Kagome layer. However, the mechanism is likely complex, similar to other materials such as YBCO[29], Fe₅₋ₓGeTe₂[30], UTe₂[31], and NdSbₓTe₂₋ₓ₋δ[32], where coupling between spatial charge modulation and magnetic order has also been reported.

## Discussion

Our experimental findings reveal an intriguing trimerization of the Fe₃Sn Kagome layer in epitaxial FeSn films that breaks the six-fold rotational symmetry but not the translational symmetry. This is in contrast to the (2 × 2) or (4 × 1) CDWs reported in AV₃Sb₅ compounds[16,17] and FeGe[18–20] that all resulted in breaking the translational symmetry. One may attribute this trimerization to a breathing

Kagome lattice characterized by anisotropic bond strengths with hopping parameters $J_A/J_B$ between nearest neighbors. Such a structure is suggested for the Fe₃Sn bilayer in Fe₃Sn₂, where the two corner-sharing triangles have different bond lengths[33]. However, the films studied here are the FeSn phase with alternatingly stacked single Sn and Fe₃Sn layers, as confirmed by the XRD data (Supplementary Fig. S2), and the observation of a perfect honeycomb without buckling on the Sn termination (Figs. 1d and 2a), in direct contrast to that observed on Sn-terminated bulk Fe₃Sn₂[3,26]. Thus, a breathing Kagome lattice in the FeSn films is unlikely.

Having ruled out the structural origin for breaking the six-fold rotational symmetry of the Kagome lattice, we discuss electronic orders, including charge and bond orders, as other possible causes. For charge density order in non-magnetic Kagome materials, while most studies focused on the impact of the van Hove singularities at the M points[7,18,20], recent work highlighted the interlayer coupling of the Kagome layers where the interactions between modes at M and L points of the BZ lead to multiple CDWs[34], including possibly the nematic order observed in CsV₃Sb₅[16]. Similarly, for magnetic Kagome material FeGe, a recent study suggested that coupling between magnetism and (2 × 2) CDW order can lead to Kekulé-like bond order in the Ge layer[35]. In the current system of epitaxial FeSn thin film, the coupling between the Sn honeycomb layer and the Fe₃Sn Kagome layer could lead to modifying the Kagome lattice. For example, if the Sn layer exhibits (√3 × √3) CDWs, such an order can cause different displacements of the Sn atoms underneath the neighboring triangles of the Kagome lattice, potentially leading to the trimerization. Evidence for such enhanced interlayer coupling can be found in the XRD data, where the (002) and (021) peaks of the FeSn are shifted slightly to larger values (Supplementary Fig. S2), indicating a smaller c-axis lattice

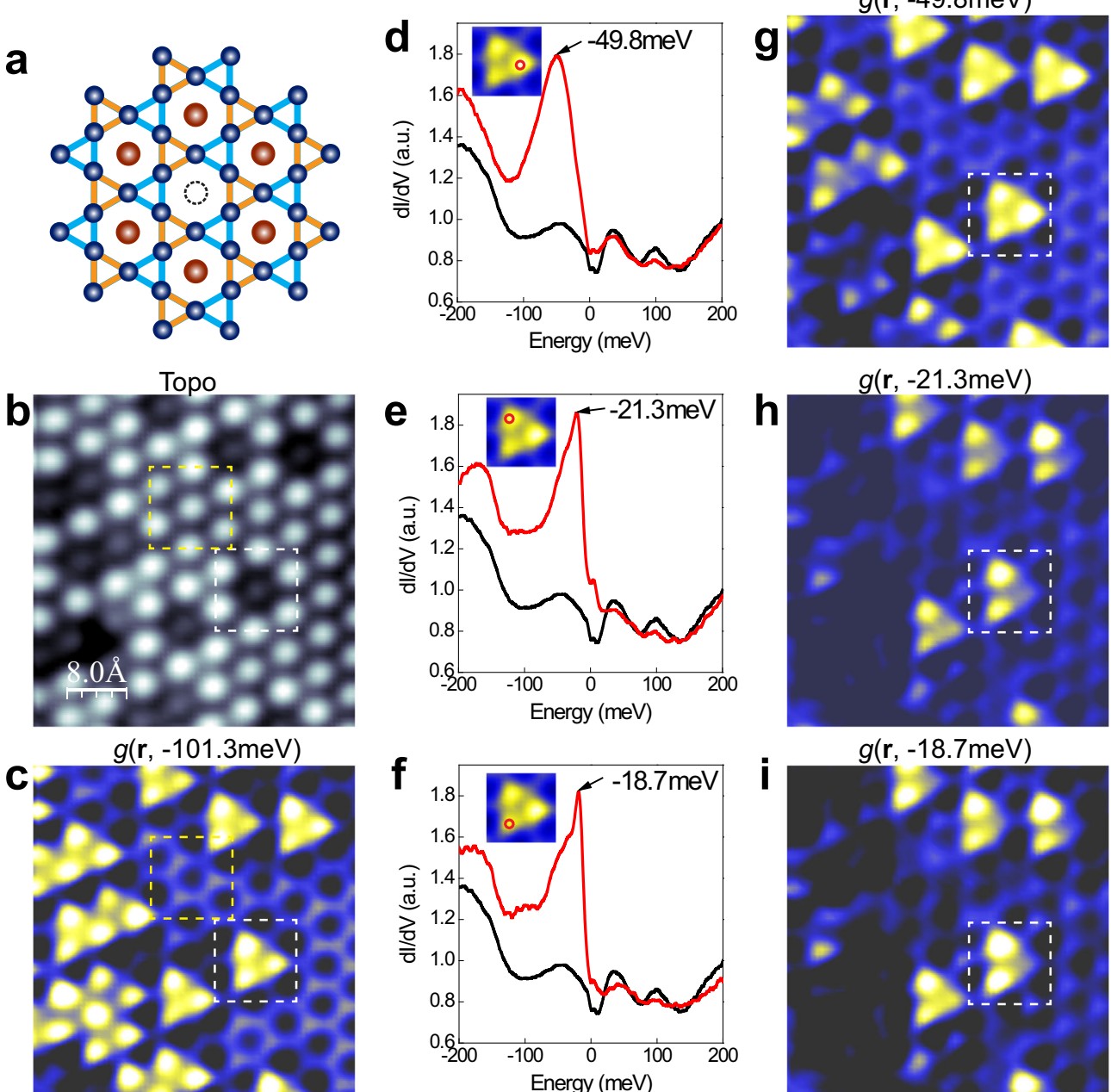

**Fig. 4 | Enhancement of trimerization by Sn vacancy induced bound states on the Fe₃Sn layer. a** Ball-and-stick model of a single Sn vacancy on the Fe₃Sn Kagome layer. **b** Topographic STM image of the Kagome layer with Sn vacancies. Setpoint: $V = 0.2$ V, $I = 5.0$ nA. The white and yellow dashed squares mark the regions with and without Sn vacancy. **c** Differential conductance map taken at −101.3 meV of the same region as (**b**), setpoint: $V = 0.2$ V, $I = 5.0$ nA, $V_{mod} = 3.0$ meV. **d**–**f** dI/dV spectra (red) measured at one of the lobes marked by the red circles and away from the defect (black). Black arrows label the energy positions of the bound states. **g**–**i** Corresponding dI/dV maps recorded at the bound state energies indicated, setpoint: $V = 0.2$ V, $I = 5.0$ nA, $V_{mod} = 3.0$ meV.

constant. Furthermore, we have reported a strain-induced substantial deformation of the Sn honeycomb on the Sn-termination for thin FeSn films[36] (Supplementary Figs. S18 and 19).

Another possible cause for breaking the six-fold rotational symmetry is the interaction-driven bond order predicted for Kagome lattices[9–11]. The trimerization is consistent with several theoretically proposed bond orders and chiral flux phases[9,10,35,37,38]. With strong bonds for the up triangles and weak bonds for the down triangles[9,10], forming such bond order waves further breaks the mirror symmetry of the Kagome lattice and opens gaps at M point[37]. While there haven't been reports of bond order in materials with the Kagome lattice, such order has been observed for the honeycomb lattice of graphene,

where it appears as alternating high/low contrasts in STM images and dI/dV maps[39–41]. Hence, the formation of bond order can also explain our observation of the trimerization of the Kagome lattice and energy-dependent high/low contrast between adjacent triangles (Fig. 3). Such bond order is expected to open a gap, consistent with our observation of the ∼80 meV gap around the $E_F$ (Fig. 1g inset), at which the contrast reversal of neighboring triangles is also observed (cf. Fig. 3f vs. j).

In addition to the trimerization of the Fe₃Sn Kagome lattice, further rotational symmetry breaking from three-fold to two-fold is also observed in defect-induced bound states (c.f., Fig. 4g–i) and energy-dependent stripe modulations in dI/dV maps (c.f., Fig. 5), suggesting nematic electronic states in FeSn[42]. The tuning of the stripe

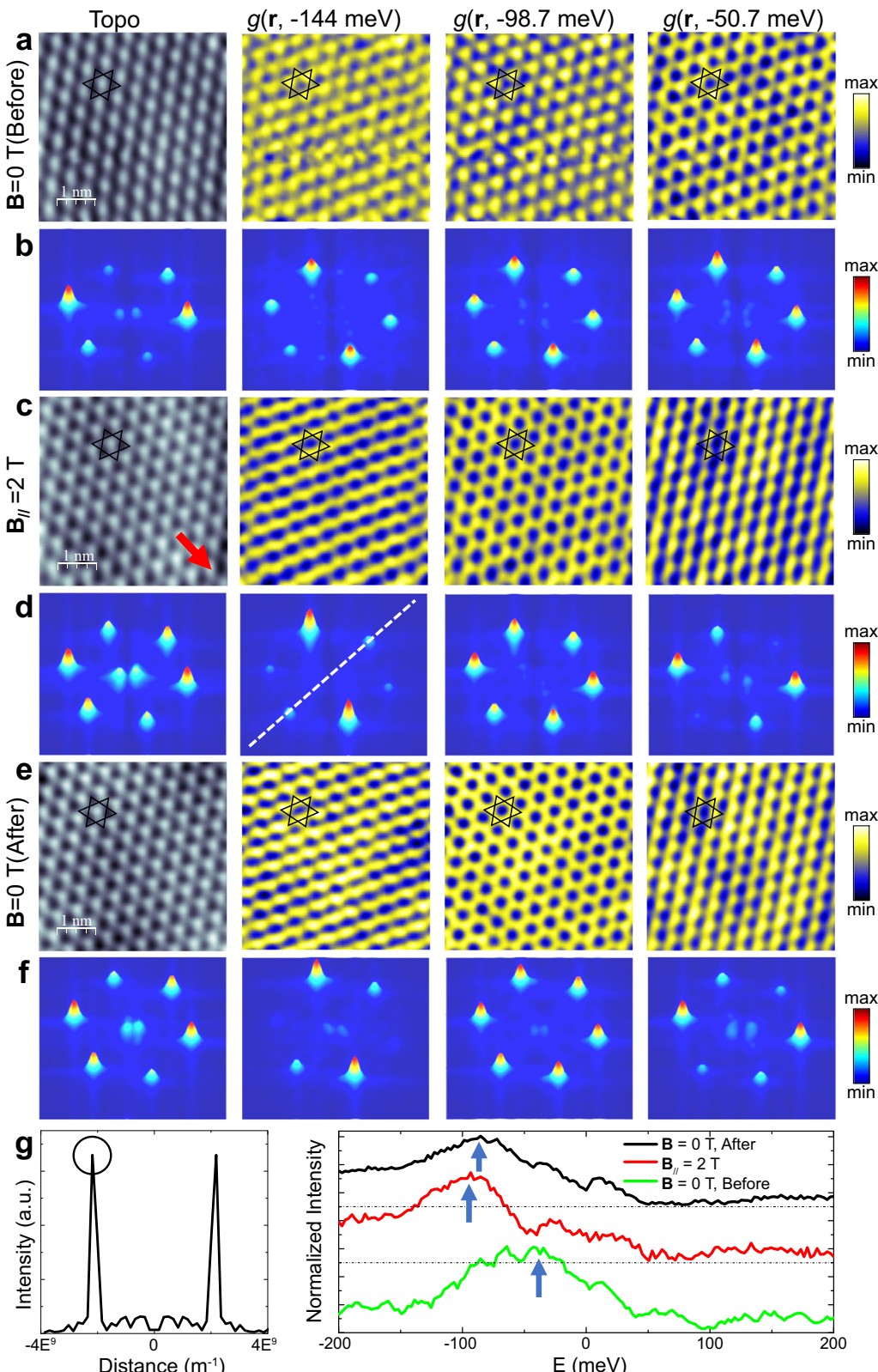

**Fig. 5 | Stripe modulations tunable by an in-plane magnetic field. a** Topographic STM image and dI/dV maps at −144, −98.7, and −50.7 meV under **B** = 0 T. Setpoint: $V$ = 0.2 V, $I$ = 0.7 nA. **b** Patterns generated by Fast Fourier transformation (FFT) of the image and maps in (**a**). **c** Topographic STM image and dI/dV maps at the same energies as (**a**) with an in-plane magnetic field **B**$_{//}$ = 2 T along the direction marked by the red arrow. Setpoint: $V$ = 0.2 V, $I$ = 0.7 nA. **d** FFT patterns generated from the image and maps in (**c**). **e** Topographic STM image and dI/dV maps at the same

energies as (a) & (c) after removing the in-plane magnetic field. Setpoint: $V$ = 0.2 V, $I$ = 0.7 nA. **f** FFT patterns generated from the image and maps in (**e**). **g** Left panel: line profile along the white dotted line in (**d**), with the FFT peak intensity marked by a black circle; and right panel: the energy-dependent FFT peak intensity for all the dI/dV maps at zero, 2 T applied field, and after the field is removed. The curves are shifted vertically for clarity. The FFT peak intensity is normalized to their corresponding averaged background intensity.

modulation by in-plane magnetic fields further indicates a strong coupling between magnetic and charge degrees of freedom, affording excellent tunability of the electronic states of FeSn. Similar magnetic field tunability has been shown on cleaved $Fe_3Sn_2$ bulk crystals[26]. The symmetry-breaking electronic orders are normally driven by correlations and will strongly depend on charge doping. Future experiments where charge doping can be tuned, e.g., depositing K atoms on the $Fe_3Sn$ surface, are called for to fully understand these symmetry-breaking phases in epitaxial FeSn films.

## Methods
### Sample preparation
The FeSn films were prepared by MBE with a Sn/Fe flux ratio >3 on Nb-doped (0.05 wt%) $SrTiO_3(111)$ substrates. The $SrTiO_3(111)$ substrates were first degassed at 600 °C for 3 h, followed by annealing at 950 °C for 1 h to obtain an atomically flat surface with step-terrace morphology. During the MBE growth, high-purity Fe (99.995%) and Se (99.9999%) were evaporated from Knudson cells on the $SrTiO_3$ substrate with temperatures between 480 and 530 °C.

### LT-STM/S characterization
The STM/S measurements were carried out in a low-temperature Unisoku STM system at $T = 4.5$ K. A polycrystalline PtIr tip was used, which was tested on Ag/Si(111) films before the STM/S measurements. dI/dV tunneling spectra were acquired using a standard lock-in technique with a small bias modulation $V_{mod}$ at 732 Hz.

### X-ray diffraction characterization
The XRD patterns of samples were obtained using a Panalytical X'Pert Pro MPD powder X-ray diffractometer with Cu $K_\alpha$ X-ray source operating at 45 kV and 40 mA in the Bragg-Brentano geometry. The spectra were collected over a 2-theta range of 30° to 50° with a solid-state X-ray detector.

### DFT calculations
First-principles calculations were carried out in the framework of generalized gradient approximation with the Perdew-Burke-Ernzerhof functionals[43] using the Vienna Ab initio simulation package (VASP). All calculations were performed with a plane-wave cutoff of 500 eV on $7 \times 7 \times 1$ Monkhorst-Pack k-point mesh. In geometric optimization, the atom positions were fully relaxed until the forces less than 0.02 eV/Å. The $Fe_3Sn$-terminated (Sn-terminated) surface was modeled by a slab geometry consisting of seven (five) atomic layers with ~15 Å of vacuum, in which the upper three layers were relaxed. The theoretical STM images were simulated using the Tersoff-Hamann approximation[44] with a larger k-points mesh ($21 \times 21 \times 1$). The STM tunneling current is proportional to the local density of states of the sample surface at the position of the tip. Therefore, the simulated STM image is the plot of the charge density distribution in a chosen energy window on one horizontal plane above the sample surface. Here, the theoretical energy window is compared to the STM bias voltage, and the vertical position of the horizontal plane is compared to the height of the STM tip. In the simulations, the orbital of the STM tip is considered an isotropic s-wave, and the density is directly obtained from the DFT calculations.

## Data availability
The data that support the findings of this study are available within the main text and Supplementary Information. Any other relevant data are available from the corresponding authors upon request.

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

## Acknowledgements

Funding for this research was provided by the U.S. Department of Energy, Office of Basic Energy Sciences, Division of Materials Sciences and Engineering under Award No. DE-SC0017632 and the US National Science Foundation under Grant No. EFMA-1741673.

## Author contributions

L.L. and H.Z. conceived and organized the study. H.Z., B.D.O., and Q.Z. performed the MBE growth and STM/S measurements. X.G. and Z.W. carried out density-functional-theory calculations. H.Z. and L.L. analyzed the data and wrote the paper. All the authors read and commented on the paper.

## Competing interests

The authors declare no competing interests.
