## [Peer Review File · Nature Communications]

Reviewers' Comments:

Reviewer #1:

Remarks to the Author:

The authors present a STM study on the electronic properties of kagome FeSn epitaxial film. They claim that they observe the first experimental observation of bond order wave (BOW). They report the following results to support their claim: a trimer state that breaks six-fold rotational symmetry on the Fe₃Sn surface, the enhancement of trimer states by Sn-vacancies, and the anomalous Zeeman shift in the tunnelling spectrum on Sn-vacancy. Given the recent experimental discovery and rapid progress on Kagome metals (e.g. the magnetic Weyl semimetal Co₃Sn₂S₂, superconducting KV₃Sb₅, Dirac and flat band in topological FeSn), the possible experimental observation of BOW phase (which has been theoretically studied in Mott systems for long time) is of great interest and suitable for the readership of Nature Communications. The rotation symmetry breaking of trimer states reported in this work are extremely interesting and intriguing; however, the evidence to support BOW as the only explanation is not convincing at this moment.

My primary concern is that the role of magnetic order at the surface is largely omitted in the discussion. In fact, the interaction within kagome magnetic metals is very complicated and it has been theoretically shown to exhibit different types of density waves (e.g. Ref. 9 or Sci. Adv. 8, eabl4108 (2022)). The authors need more evidence to support such a strong claim. Secondly, the conductance maps showing trimer features were taken in a very small field of view (only unit cells). The spatial variation in dI/dV are quite pronounced in Figure 3 (yet no defect is visible in the topographic image). It is desirable to show the conductance map with larger field of view and also corresponding FFT analysis, which can also rule out the possible effect from Sn-vacancy below the surface. Several issues should also be addressed which I will discuss below.

3. What is the temperature for STM measurements?? I could not find this essential info anywhere.

4. To minimize the surface energy due to the polar STO(111) surface, different types of surface reconstruction can occur, depending on the treatment process, and stoichiometry. The termination can also greatly affect the electronic behaviours at the interface (e.g. LAO/STO system). Domain structure with different reconstructions (but without atomic resolution) is shown in Figure S1e. What are the surface structure of STO(111)? What's the termination of STO(111)? It may be responsible for the orientation dependent trimer or dimer in Figure 5 since the structure underneath can be very different. What are those bright clusters on the STO(111) surface?

5. In Figure 5, not all Sn-vacancies exhibit enhanced trimer state in both $g(r, -100\text{meV})$ and $g(r, -16\text{meV})$ maps. Can the author comment?

6. In Figure S2, the (002) peak is not visible (different from Ref. 5 and Ref. 22). Even though the STM images show identical atomic structure as Sn and Fe₃Sn termination, the size of STM images are only few nanometres. It will be more convincing to show images with large area (e.g. Figure S11a or 11b) in the main text.

7. The DFT simulated images (Figure S5, S6 and S8) are used to support the STM data. However, the DFT didn't catch the difference in dI/dV between the A and B sites or the dimer state in Fe₃Sn-layer. Perhaps the authors can comment on the possible reason.

8. The B-dependent tunnelling spectra were taken at A or B site. How about the B-dependent spectra taken at the other site?

9. Can the author include the spectra taken between -1T and 1T for Figure 6c and 6d?

10. I don't understand how the observation of trimer state with 6nm-thick film can rule out the strain effect in the discussion. May the authors elaborate this point?

11. Line 219: The gap opening in Fe₃Sn layer may be due to BOW but it can also be other reasons. For example, the gap opening in Sn layer is "unknown".

Other minor points and corrections:

Figure 1c should be enlarged.

The energy in Figure 3d and 3e are the same.

Line 67: Ref. 6 doesn't report the FeSn film on STO(111).

Figure 6b and Figure S11g are not consistent?

In the last paragraph in p.7, the authors stated "we also observe a novel two-fold oscillatory response to in-plane magnetic field". However, this behaviour is already reported in Fe₃Sn single crystals (Ref. 25) but with a "phase shift" to FeSn.

The g-factor is 42.5, compared with 210 in Fe₃Sn single crystal (Ref. 25). Can the author comment?

Typo: Figure 6c, dI/dV spectra taken at the RED dot in b.

Typo: Figure S15c and S15d: the peak position should be in meV?

Reviewer #2:

Remarks to the Author:

Huimin Zhang et al. report an MBE growth of Kagome magnet FeSn on STO substrates, and on top of it, they performed STM/STS and magnetic field dependence study. On the Fe₃Sn terminated surface, they observed an 80 meV gap around the Fermi level. Both the bias-dependent atomic image and differential conductance image exhibit three-fold symmetry, from which they claimed to be bond-order wave patterns, and the 80 meV gap is attributed to the bond-order wave gap. In addition, they also studied local Sn vacancy defects and the associated defect bond states under magnetic fields, in which they observed an energy shift in the bond states as a function of magnetic field strength and rotation angle. Finally, they proposed two possible bond order pictures and argued that bond order's formation origin is electron-electron interactions.

The Kagome magnets is currently a very interesting and active topic. The growth of FeSn using MBE is not easy and can potentially enable new tuning knobs in the thin film regime of Kagome magnets. The STM/STS data reported here is also extensive. However, their interpretation seems to be a bit bold and not convincing. I cannot recommend a publication on Nature Communications based on a few points below:

(1) The author claimed a successful growth of FeSn flakes with two types of termination. However, most flakes have a flat top, and their thickness does not seem to be integer multiples of unit cell height, 4.4 Å. It would be a lot more convincing to achieve layer-by-layer growth of FeSn to ensure no other chemical compositions are mixed within the flake. For example, one can have a topographic cross-section from the substrate to the top of the flake showing a gradual build-up of the layers, each of which exhibits a correct thickness.

(2) Another big concern is the attribution of atomic registry on Fe₃Sn terminated surface. According to a few other studies on similar systems [Nat. Mater. 20, 1353-1357 (2021) arXiv:2203.01888, arXiv: 2203.08077], the central Sn atom in the David Star is usually not visible and appears as dark rather than bright. Can the author comment on this difference? In other words, how did the author determine the atomic lattice correspondence on the STM image? In addition, the bias-dependent atomic images presented in Fig. 2 seem to have some drift and would be difficult to compare across different images. This is a very crucial point that needs to be clarified carefully because the main argument in the manuscript is based on the atomic lattice symmetry analysis.

(3) How could the author differentiate a bond order wave from a charge order wave? Both the bond-order wave and charge-order wave are electronic behaviors, and a charge-order wave can also give rise to a gap at the Fermi level. A study [arXiv:2203.01888] on an isostructural system FeGe claims such a Fermi level gap is associated with a charge order gap.

(4) Has the author tried to raise the temperature to be above the bond-order transition temperature and observe a disappearance of the gap and recovery of the six-fold symmetry?

(5) Impurities such as vacancies can incur a redistribution of the local chemical bonds and naturally break the original atomic lattice symmetry [Phys. Rev. Lett. 123, 076801]. I do not quite

understand why the defect-induced symmetry breaking is significant here and how to relate it to a defect-free case. Also, what are the behaviors for other types of local defects compared to the Sn vacancy defect?

A few other minor comments:

(6) Were the dI/dV maps performed in a constant-height or constant-current mode. Usually, the constant-height dI/dV mode can reduce the atomic topographic corrugation and can reflect the charge distribution more realistically.

(7) What is the STM/S study temperature? This information can help evaluate the critical "bond-order" transition temperature.

(8) The manuscript text can be further improved to make it easier to follow. Manuscript caption, for example, in lines #405 and #410, needs to be carefully checked to avoid mistakes.

REVIEWER COMMENTS

Reviewer #1 (Remarks to the Author):

The authors present a STM study on the electronic properties of kagome FeSn epitaxial film. They claim that they observe the first experimental observation of bond order wave (BOW). They report the following results to support their claim: a trimer state that breaks six-fold rotational symmetry on the Fe₃Sn surface, the enhancement of trimer states by Sn-vacancies, and the anomalous Zeeman shift in the tunnelling spectrum on Sn-vacancy. Given the recent experimental discovery and rapid progress on Kagome metals (e.g. the magnetic Weyl semimetal Co₃Sn₂S₂, superconducting KV₃Sb₅, Dirac and flat band in topological FeSn), the possible experimental observation of BOW phase (which has been theoretically studied in Mott systems for long time) is of great interest and suitable for the readership of Nature Communications. The rotation symmetry breaking of trimer states reported in this work are extremely interesting and intriguing; however, the evidence to support BOW as the only explanation is not convincing at this moment.

We thank the reviewer for the positive comments on our work and have conducted additional experiments to address the main concern below.

My primary concern is that the role of magnetic order at the surface is largely omitted in the discussion. In fact, the interaction within kagome magnetic metals is very complicated and it has been theoretically shown to exhibit different types of density waves (e.g. Ref. 9 or Sci.Adv. 8, eabl4108 (2022)). The authors need more evidence to support such a strong claim.

This is a very valid concern. Indeed, for FeSn, the Fe in the Kagome lattice is ferromagnetically ordered, while the couple between the layers are antiferromagnetic (*J. Phys. Soc. Jpn.* **22**, 1210 (1967), *J. Phys. F: Met. Phys.* **11**, 281 (1981)). In this work, however, non-spin-polarized tips are used for STM imaging and spectroscopy; hence magnetic information is not expected directly. In

the original manuscript, we have inferred such information based on the anomalous response of Sn-vacancy bound states to the magnetic field applied in both out-of- and in-plane directions.

To address the role of magnetic order at the surface more directly, we have systematically investigated the trimer states' response to an applied magnetic field in both out-of- and in-plane directions. We find that while the out-of-plane field has minimum impact, the three-fold symmetry of the trimer structure is reduced to two-fold at specific energies under in-plane magnetic field, clearly supporting the formation of bond-order waves. As these results provide more direct evidence for bond order waves, we have replaced the original Fig. 6 in the revised manuscript with the following discussion on pages 6-7.

“Breaking the three-fold symmetry of the trimer structure by an in-plane magnetic field. Given that Fe atoms in the kagome layer are ordered ferromagnetically, we now examine the impact of the magnetic field on the trimerization. Note that non-spin-polarized tips are used for STM imaging and spectroscopy, magnetic information is not expected directly. Nevertheless, we observe breaking the three-fold symmetry of the trimer structure by an in-plane magnetic field of 1-2 T. Figure 6a shows an STM topography image and corresponding dI/dV maps taken at the energies specified at zero magnetic field, where a close-packed structure is observed. The close-packed structure is further confirmed by the FFT patterns shown in Fig. 6b, albeit there exist slight variations of the FFT peak intensities. The topography image remains unchanged with an applied 2 T in-plane magnetic field (Fig. 6c left panel). Remarkably, stripe modulations are observed in the dI/dV maps at the same energies. At -144 meV, the close-packed structure is modulated along the direction marked by the orange arrow, leading to substantial enhanced intensity at two of the peaks in the FFT pattern (Fig. 6d). Interestingly, a honeycomb structure is observed at -98.7 meV, leading to a symmetrical FFT pattern. Then, the structure reverts to be close packed again at -50.7 meV, but with a stripe modulation 60° from that at -144 meV. This stripe formation is also clear in the FFT patterns shown in Fig. 6d (right panel). These observations indicate a magnetic field induced symmetry breaking of the trimer structure, forming stripes, but only along two directions 60° apart. Additional data is provided in the Supplementary Information Figs. S14-16 to support this observation. Note that there are small modulations in data taken with a 9 T out-of-plane magnetic field, but no clear stripe formation is observed. Overall, the in-plane

magnetic field induces a selective dimerization that forms a stronger bond at a specific energy, as shown in Fig. 6e, second panel. At different energy, a different dimerization leads to stripes along a different direction, e.g., as those shown in Fig. 6e, last panel. If all the three bonds are dimerized, a honeycomb structure could form, as shown in the middle panel in Fig. 6e. This reduction of the three-fold symmetry of the trimer structure to two-fold stripes with an applied in-plane magnetic field indicates a strong coupling between magnetic and charge degrees of freedom, affording great tunability of the ground states of FeSn. Similar magnetic field tunability has been shown on the surface of cleaved Fe₃Sn₂ bulk crystals²⁵.”

As such, our experimental findings of trimerization of the Fe₃Sn kagome layer and magnetic field-induced dimerization in epitaxial FeSn films break the six-fold rotational/mirror symmetry, but not the translational symmetry. This is in direct contrast to the (2 x 2) or (4 x 1) CDWs reported in AV₃Sb₅ compounds and FeGe that all resulted in the breaking of the translational symmetry. Therefore, we conclude that the formation of CDW is unlikely the origin of our observations. This discussion has now been added to the revised manuscript on page 7.

We also thank the reviewer for suggesting the paper on the possible smectic order and have added it as a new Ref. 31.

Secondly, the conductance maps showing trimer features were taken in a very small field of view (only unit cells). The spatial variation in dI/dV are quite pronounced in Figure 3 (yet no defect is visible in the topographic image). It is desirable to show the conductance map with larger field of view and also corresponding FFT analysis, which can also rule out the possible effect from Sn-vacancy below the surface.

A larger-scale STM image and the corresponding dI/dV maps are shown below in Fig. R1. The FFTs of the dI/dV maps are shown in Figs. R1f-j and Figs. R1p-t. Visual inspection of the FFT maps doesn't reveal additional features that might be from Sn-vacancy below the surface.

Fig. R1. Differential conductance map of the kagome lattice on the Fe_3Sn layer. **a**, Topographic STM image of the Fe_3Sn layer. Setpoint: $V = 0.15$ V, $I = 5.0$ nA. **b-t**, Differential conductance maps of the same location in **a** at the specified energies. Setpoint: $V = 0.15$ V, $I = 5.0$ nA, $V_{mod} = 3$ meV.

Several issues should also be addressed which I will discuss below.

3. What is the temperature for STM measurements?? I could not find this essential info anywhere.

Reply: The STM measurements were carried out at the temperature $T = 4.5$ K. We have added the information in Methods on Page 9.

4. To minimize the surface energy due to the polar STO(111) surface, different types of surface

reconstruction can occur, depending on the treatment process, and stoichiometry. The termination can also greatly affect the electronic behaviours at the interface (e.g. LAO/STO system). Domain structure with different reconstructions (but without atomic resolution) is shown in Figure S1e. What are the surface structure of STO(111)? What's the termination of STO(111)? It may be responsible for the orientation dependent trimer or dimer in Figure 5 since the structure underneath can be very different. What are those bright clusters on the STO(111) surface?

Fig. R2. Topographic STM images of SrTiO₃(111). Step-terrace morphology in **a** and a (4 × 4) reconstruction in **b**. Setpoint: $V = 3.0$ V, $I = 30$ pA (**a**) and $V = 1.0$ V, $I = 500$ pA (**b**).

Atomic resolution imaging reveals a (4 × 4) reconstruction, as shown in Fig. R2**b**, indicating TiO₂ excess on the surface (*Phys. Rev. Lett.* **114**, 226101 (2015)). This is consistent with earlier reports on epitaxial growth of FeSn/SrTiO₃(111), where a Ti-rich region is found at the interface (*Nat. Commun.* **12**, 5345 (2021)). Therefore, the bright clusters on the STO(111) surface are probably SrO clusters since the surface usually is SrO deficient.

Yes, the orientation dependence is likely due to a different epitaxial relationship with the STO substrate, as we state on Page 6, “likely due to slightly different epitaxial relationship with the STO substrate that exhibits different domains (Supplementary Figure S1)”.

5. In Figure 5, not all Sn-vacancies exhibit enhanced trimer state in both $g(r, -100\text{meV})$ and $g(r, -16\text{meV})$ maps. Can the author comment?

This is likely due to a strong coupling between the neighboring bound states. As shown in Figs. R3a-c, for an isolated Sn vacancy, the trimer states become a dimer in $g(r, -16\text{meV})$. The dimer states are not impacted for two vacancies that are slightly separated (Figs. R3d-f). In contrast, the bound states can almost disappear completely for two or three neighboring vacancies (Figs. R3g-i). We have included this in the Supplementary Fig. S13, and included the following discussion in the revised manuscript on Page 6: “Note that not all of the Sn-vacancies exhibit enhanced trimer states in dI/dV maps, e.g., $g(r, 16\text{ meV})$, which is due to strong coupling between the neighboring Sn vacancies, as shown in Supplementary Figure S13.”.

Fig. R3. Bound states of Sn vacancy clusters at the Kagome layer. a-c, Topographic STM image and dI/dV maps of single Sn vacancy. Setpoint: $V = 0.15\text{ V}$, $I = 5.0\text{ nA}$, $V_{mod} = 3.0\text{ meV}$. d-f, Topographic STM image and dI/dV maps of two separate Sn vacancies. Setpoint: $V = 0.2\text{ V}$, $I = 5.0\text{ nA}$, $V_{mod} = 3.0\text{ meV}$. g-i, Topographic STM image and dI/dV maps of two Sn vacancies close to each other. Setpoint: $V = 0.2\text{ V}$, $I = 5.0\text{ nA}$, $V_{mod} = 3.0\text{ meV}$. j-l, Topographic STM image and dI/dV maps of three nearby Sn vacancies. Setpoint: $V = 0.15\text{ V}$, $I = 5.0\text{ nA}$, $V_{mod} = 3.0\text{ meV}$.

6. In Figure S2, the (002) peak is not visible (different from Ref. 5 and Ref. 22). Even though the STM images show identical atomic structure as Sn and Fe₃Sn termination, the size of STM images are only few nanometres. It will be more convincing to show images with large area (e.g. Figure S11a or 11b) in the main text.

During the period of revising the manuscript, we constructed another MBE system for the growth FeSn films. While the growth mode is still island growth, their sizes are larger (> 100 nm). The XRD results on one of the new films are included in the new Supplementary Fig. S2 (copied below). In addition to the (021) peak, a more pronounced (002) peak is also seen. This further confirms STM/S results are indeed on the surface of FeSn films.

7. The DFT simulated images (Figure S5, S6 and S8) are used to support the STM data. However, the DFT didn't catch the difference in dI/dV between the A and B sites or the dimer state in Fe₃Sn-layer. Perhaps the authors can comment on the possible reason.

The trimerization, thus the formation of bond order waves, is likely the result of strong e-e interactions. Such effects are not included in our standard DFT calculations, which can't catch the difference in the dI/dV tunneling spectra between different sites, and dimer states in the Kagome layer.

8. The B-dependent tunnelling spectra were taken at A or B site. How about the B-dependent spectra taken at the other site?

The B-dependent tunneling spectra taken at other sites are shown in Fig. R4.

Fig. R4. No obvious oscillatory behavior under in-plane magnetic field in the region away from defects. a, Topographic STM image of Fe₃Sn termination, setpoint: $V = 0.4$ V, $I = 1.0$ nA. **b,** dI/dV

spectra under in-plane magnetic fields with cyan and red arrow mark two peak positions. **c,d** Shift of peak position as a function in-plane magnetic field azimuth angle.

9. Can the author include the spectra taken between -1T and 1T for Figure 6c and 6d?

The spectra are shown below in Fig. R5. However, as discussed above, we have replaced Fig. 6 with a new figure to highlight the additional symmetry breaking under the applied in-plane magnetic field. Thus, this set of data is not included in the revised manuscript.

Fig. R5. dI/dV spectra as a function of out-of-plane magnetic fields B_{\perp} from -1T to 1T indicated. The cyan arrows mark the peak position of the bound state.

10. I don't understand how the observation of trimer state with 6nm-thick film can rule out the strain effect in the discussion. May the authors elaborate this point?

We have studied the effect of strain systematically, and the results will be published elsewhere. Briefly, we find that the Sn-honeycomb layer is easily deformed forming a buckled structure, leading to rows that break the rotational symmetry of the honeycomb. This can be clearly seen in the two STM images taken on unstrained and strained films, with the strain mostly determined by the film thickness. In general, thinner films are strained, while thicker ones are not. In contrast, the Kagome is mostly immune from distortion by the same amount of strain. In this work, we mainly determine the status of the strain, by simply the observation of perfect honeycomb vs. distorted honeycomb with *in-situ* STM imaging. In the future, *ex-situ* XRD measurements can be carried out to quantify the amount of strain.

As shown in Supplementary Fig. S17, on the same island with both Sn and Kagome termination, the Sn-honeycomb is strongly distorted with two types of rows, while the trimer structure of the Kagome layer remains intact.

11. Line 219: The gap opening in Fe₃Sn layer may be due to BOW but it can also be other reasons. For example, the gap opening in Sn layer is “unknown”.

The gap opening in Sn layer can be attributed to multiple reasons, one of which is electron interactions commonly reported in 2D topological insulators. For example, a similar gap is observed in 1T'-WTe₂ (*Nat. Commun.* **9**, 4071 (2018)) and 1T'-WSe₂ (*Nat. Commun.* **9**, 3401 (2018)), which was attributed to the opening of the Coulomb gap always pinned at the Fermi level.

We have added the following in the revised manuscript on Page 3: “The nature of this gap is unknown, which could be attributed electron interactions similar to that reported in 2D topological insulators 1T'-WTe₂²⁶.”

Other minor points and corrections:

Figure 1c should be enlarged,

We thank the reviewer for the suggestion and have updated Figure 1.

The energy in Figure 3d and 3e are the same,

We thank the reviewer for pointing it out and have revised the figure caption.

Line 67: Ref. 6 doesn't report the FeSn film on STO(111),

We thank the reviewer for pointing it out and have corrected the references.

Figure 6b and Figure S11g are not consistent?

We thank the reviewer for pointing out the mislabeling and have corrected the Figure S11g (Fig. S9g in updated Supplementary Information).

In the last paragraph in p.7, the authors stated “we also observe a novel two-fold oscillatory response to in-plane magnetic field”. However, this behaviour is already reported in Fe₃Sn single crystals (Ref. 25) but with a “phase shift” to FeSn. The g-factor is 42.5, compared with 210 in Fe₃Sn single crystal (Ref. 25). Can the author comment?

Typo: Figure 6c, dI/dV spectra taken at the RED dot in b,

As discussed above, we have replaced Fig. 6 with a new figure to highlight the additional symmetry breaking under the applied in-plane magnetic field.

Typo: Figure S15c and S15d: the peak position should be in meV?

We thank the reviewer for pointing out the mislabeling and have corrected them in the revised manuscript.

Reviewer #2 (Remarks to the Author):

Huimin Zhang et al. report an MBE growth of Kagome magnet FeSn on STO substrates, and on top of it, they performed STM/STS and magnetic field dependence study. On the Fe₃Sn terminated surface, they observed an 80 meV gap around the Fermi level. Both the bias-dependent atomic image and differential conductance image exhibit three-fold symmetry, from which they claimed to be bond-order wave patterns, and the 80 meV gap is attributed to the bond-order wave gap. In addition, they also studied local Sn vacancy defects and the associated defect bond states under magnetic fields, in which they observed an energy shift in the bond states as a function of magnetic field strength and rotation angle. Finally, they proposed two possible bond order pictures and argued that bond order's formation origin is electron-electron interactions.

The Kagome magnets is currently a very interesting and active topic. The growth of FeSn using MBE is not easy and can potentially enable new tuning knobs in the thin film regime of Kagome magnets. The STM/STS data reported here is also extensive. However, their interpretation seems to be a bit bold and not convincing. I cannot recommend a publication on Nature Communications based on a few points below:

(1) The author claimed a successful growth of FeSn flakes with two types of termination. However, most flakes have a flat top, and their thickness does not seem to be integer multiples of unit cell height, 4.4 Å. It would be a lot more convincing to achieve layer-by-layer growth of FeSn to ensure no other chemical compositions are mixed within the flake. For example, one can have a topographic cross-section from the substrate to the top of the flake showing a gradual build-up of the layers, each of which exhibits a correct thickness.

What the reviewer is referring to is a layer-by-layer growth, which unfortunately is not the case for FeSn growth on STO(001) substrate. The growth mode is found to be island growth, which is intrinsically determined by the strain caused by lattice mismatch and the surface energy difference between the film and STO substrates. Our latest work shows that larger domains can be grown, but the growth mode is still island growth (as shown in Supplementary Fig. S2).

Nevertheless, steps are observed on the islands, e.g., the one marked by the red square, the step height is 0.42 nm, consistent with the expected one atomic layer thickness of FeSn (0.45 nm).

Fig. R6. Interlayer atomic spacing. **a**, STM image in derivative mode, showing an island growth mode. Setpoint: $V = 3$ V, $I = 10$ pA. **b**, Line profile across an island highlighted by a red square in (a), showing a step height of 0.42 nm, consistent with the one atomic layer thickness of FeSn (0.45 nm).

(2) Another big concern is the attribution of atomic registry on Fe₃Sn terminated surface. According to a few other studies on similar systems [Nat. Mater. 20, 1353-1357 (2021) arXiv:2203.01888, arXiv: 2203.08077], the central Sn atom in the David Star is usually not visible and appears as dark rather than bright. Can the author comment on this difference? In other words, how did the author determine the atomic lattice correspondence on the STM image? In

addition, the bias-dependent atomic images presented in Fig. 2 seem to have some drift and would be difficult to compare across different images. This is a very crucial point that needs to be clarified carefully because the main argument in the manuscript is based on the atomic lattice symmetry analysis.

Most prior studies are on the surface of cleaved bulk materials. In contrast, our samples are on the surface of films grown by MBE under Sn-rich conditions (Sn/Fe flux ratio > 3). As such, there is likely the Kagome layer is also with the central Sn atom. We have included the following discussion on pages 4-5 in the revised manuscript: “We note that there are recent studies on similar systems where the central Sn/Ge atom is usually not visible and appears as low contrast rather than bright in the STM images^{16,27,28}. However, these prior studies are on the surface of cleaved bulk materials. In contrast, our samples are on the surface of films grown by MBE under Sn-rich conditions (Sn/Fe flux ratio > 3), likely leading to a different atomic registry on the Fe₃Sn terminated surface.”.

The reviewer is correct that the bias-dependent images show drafts from one frame to the next. However, we can use the existence of a Sn-vacancy as a reference to assign the atomic lattice, as shown in Fig. S5 (copied below as Fig. R7). We have updated the Fig. 2 and caption in the revised manuscript.

Fig. R7. Bias dependent STM imaging of the Fe_3Sn kagome lattice. STM images of the Fe_3Sn -termination at the bias voltage specified. Setpoint: $I = 3.0$ nA. The ball-and-stick model of the kagome lattice is overlaid on the surface.

(3) How could the author differentiate a bond order wave from a charge order wave? Both the bond-order wave and charge-order wave are electronic behaviors, and a charge-order wave can also give rise to a gap at the Fermi level. A study [arXiv:2203.01888] on an isostructural system FeGe claims such a Fermi level gap is associated with a charge order gap.

Bond order wave (BOW) is mostly driven by e-e interactions and has been studied for Mott systems theoretically. It has been shown to exist in a 1D extended Hubbard model at half-filling in addition to CDW and SDW. The difference between the two types of charge orders is clearly seen in the calculated phase diagram (*Phys. Rev. B* **65**, 155113 (2002)). However, CDWs and BOWs can coexist in 2D, depending on filling factors (*Phys. Rev. B* **62**, 13400 (2000)). We agree it's challenging to experimentally differentiate the two types of charge orders by STM imaging alone.

In the original manuscript, we have inferred their difference based on the anomalous response of Sn-vacancy bound states to the magnetic field applied in both out-of- and in-plane directions.

To differentiate the two types of orders more definitely, we have carried out systematic imaging and spectroscopy of the FeSn films probe the impact of magnetic order on the bond order waves under the applied magnetic field. The results indicate that while the out-of-plane field has minimum impact, the three-fold symmetry of the trimer structure is further reduced to two-fold at specific energies under in-plane magnetic field. The results are now presented in new Fig. 6 (copied below). Two types of stripes are observed at -144 meV and -50.7 meV, which are absent without the applied magnetic field. This indicates that the trimer order is further coupled to the magnetic order of the Fe, clearly supporting the formation of bond-order waves. Additional maps

at other energies, and with a 9 T magnetic field applied out-of-plane direction are provided in Supplementary Figs. S14-16, all supporting a magnetic field tunable bond order.

As such, our experimental findings of trimerization of the Fe_3Sn kagome layer and magnetic field induced dimerization in epitaxial FeSn films breaks the six-fold rotational/mirror symmetry, but not the translational symmetry. This is in direct contrast to the (2×2) or (4×1) CDWs reported in AV_3Sb_5 compounds and FeGe that all resulted in the breaking of the translational symmetry. Therefore, we conclude that the formation of CDW is unlikely the origin of our observations. This discussion has now been added to the revised manuscript on pages 7-8.

(4) Has the author tried to raise the temperature to be above the bond-order transition temperature and observe a disappearance of the gap and recovery of the six-fold symmetry?

This is a good suggestion, but such an experiment was not performed. The observed gap is > 80 meV, much larger than 25 meV for room temperature. So, we do not expect to see the closing of the gap.

(5) Impurities such as vacancies can incur a redistribution of the local chemical bonds and naturally break the original atomic lattice symmetry [Phys. Rev. Lett. 123, 076801]. I do not quite understand why the defect-induced symmetry breaking is significant here and how to relate it to a defect-free case. Also, what are the behaviors for other types of local defects compared to the Sn vacancy defect?

On the Sn-termination, there are two types of defects, one is the Sn di-vacancy, and the other a substitutional defect, as shown below:

Fig. R8. Two types of defects observed on the Sn-termination in FeSn/STO(111) films. a, Atomic resolution image of the Sn termination (S) with a honeycomb lattice. Setpoint: $V = 0.2$ V, $I = 3.0$ nA. Three-fold and two-fold defects (dashed white circle) are observed. **b,** Sn bi-vacancy in S layer labeled by the dashed white circle in **a**, setpoint: $V = 2.0$ V, $I = 10$ pA. **c,** Three-fold defect centering at the triangle of S layer, setpoint: $V = 2.0$ V, $I = 10$ pA. **d-e,** dI/dV spectra measured at the cyan dots in **b** and **c**, respectively.

The Sn di-vacancy defect induced bound states, which are clearly two-fold symmetrical, reflecting its structural characteristics. However, the bound states of the substitutional defect are three-fold, again reflecting its structural characteristics.

On the Kagome layer, the most common defect is the Sn-vacancy due to the removal of a Sn atom, which doesn't break the six-fold symmetry of the Kagome lattice. However, the trimerization is much enhanced around the vacancy site. Furthermore, such enhancement exhibits a two-fold symmetry at certain energies, as shown in Figs. 4&5. Such information was used to support the additional symmetry breaking in the electronic structure of the Kagome layer.

A few other minor comments:

(6) Were the dI/dV maps performed in a constant-height or constant-current mode. Usually, the constant-height dI/dV mode can reduce the atomic topographic corrugation and can reflect the charge distribution more realistically.

The dI/dV maps were performed in the constant-current mode. We have also carried dI/dV spectroscopy using different sets of tunneling currents at a fixed bias voltage. The results (copied below as Fig. R9) show that while the magnitude of the spectra increases, the shape remains the same. We have included this in Supplementary Fig. S3.

Fig. R9. Setpoint-dependent dI/dV spectra. Atomic resolution image (a), and dI/dV tunneling spectra taken at three typical locations (b-d) (Setpoint: $V = 200$ mV, $I = 0.3$ nA to 2.5 nA).

(7) What is the STM/S study temperature? This information can help evaluate the critical “bond-order” transition temperature.

The STM measurements are carried out at $T = 4.5$ K. We have added the temperature in Methods on Page 9.

(8) The manuscript text can be further improved to make it easier to follow. Manuscript caption, for example, in lines #405 and #410, needs to be carefully checked to avoid mistakes.

We thank the reviewer for the comment and have improved the text in the revised manuscript.

Reviewers' Comments:

Reviewer #1:

Remarks to the Author:

In this revision, the authors modified the manuscript and included more data, particularly magnetic field dependent conductance maps in figure 6. While I appreciate the efforts but I am still not convinced the data supports their main conclusion, i.e. the bond order wave (BOW). As the authors mentioned in their rebuttal, it's challenging to distinguish BOW from CDW. The dependence of magnetic field only proves the coupling between the spatial charge modulation and the magnetic order, which has also been observed in YBCO (e.g. NatComms 7, 11494), Fe₅GeTe₂ (e.g. PRB104.165101), UTe₂ (2207.09491), NdSbTe (2301.13102) and others, but not the direct evidence of BOW. In addition, the data taken at zero magnetic field in figure 6 already show the broken rotational symmetry from the FFT analysis in all energies. One concern is if this could be an extrinsic effect, such as strain from substrates or the tip shape. Then, the in-plane magnetic field may enhance the modulation along one direction. However, it is not possible to tell from the data when the color scale is only marked as high/low and without knowing the direction of in-plane magnetic field in these images.

Overall, these results do demonstrate interesting symmetry breaking phenomena and spatial charge modulation, which appears to be signatures of kagome materials. If the authors remove BOW from the title, tone down their claims and include other possible origins (e.g. CDW or van Hove singularity that have also been observed in other kagome compounds), I will consider to recommend its publication in Nature Communications.

Reviewer #2:

Remarks to the Author:

The author has performed additional experiments on magnetic field dependence of the density wave order and observed a three-fold symmetry-breaking stripe order phase using scanning tunneling microscopy. The MBE growth result and the scanning tunneling microscopy observed symmetry-breaking density waves are interesting and worth attention. However, the claim on forming bond order waves (BOW) is not convincing. Thus, I can only make recommendations after the questions below are properly addressed.

1. The BOW phase is characterized by alternating strengths of the expectation value of the kinetic-energy operator on the bonds, in other words, hopping parameters [Physical Review B, 65, 155113 (2002)], and the charge density should be uniform [Physical Review B 82, 075125 (2010)]. The requirement of the bond wave order is quite critical and only happens at a tiny region in the phase space at specific fractional filling like 1/3 or 1/4. What is the filling factor here? STM measures mostly charge densities; even if a bond order exists, I am not convinced that STM can tell bond order waves from charge density waves. To my knowledge, the bond order wave has yet to be observed experimentally.

2. In Fig. S6f, DFT simulated STM image does not capture the 6-fold symmetry breaking as is shown in the experimental STM image. Can the author comment on it? Can the author provide more detail on the DFT simulation?

3. Why would the bond order depend on the bias? Why do the strong bond and the weak bond switch from up-triangle-site to down-triangle-site position as a function of energy?

4. I do not understand the assignment of the lattice registry in Fig. 5. As the lattice has 6-fold symmetry and Fig. 5 presents three separate islands, 60 degrees rotation should be equivalent for them. In other words, the relative angle from Fig. 5a and 5d can also be 69.2 degrees. Then rotating Fig. 5e and 5f clockwise by 60 degrees, their trimerization and dimerization direction would be the same as Fig. 5b and 5c, respectively. The same applied to Fig. 5g to 5i. Then the argument in electronic rather than structural structure does not work (line #160 to line #163).

5. Applying an in-plane magnetic field naturally breaks the 3-fold symmetry into two-fold as the B field can only be applied along one direction. What is the relative angle between the applied B field

and the formed stripe orientation? Single q smectic bond order mentioned in Ref. #31 does not rely on the magnetic field. This piece of the experiments is interesting and worth reporting but I do not think it becomes additional evidence for the bond order formation.

Overall, the claim on bond order wave is not convincing. The author should re-edit the title and the manuscript to focus on reporting the interesting discovery of symmetry-breaking energy and magnetic-field dependent density waves (electronic order) rather than making a big but not convincing claim on bond order waves.

Reviewer #1 (Remarks to the Author):

In this revision, the authors modified the manuscript and included more data, particularly magnetic field dependent conductance maps in figure 6. While I appreciate the efforts but I am still not convinced the data supports their main conclusion, i.e. the bond order wave (BOW). As the authors mentioned in their rebuttal, it's challenging to distinguish BOW from CDW. The dependence of magnetic field only proves the coupling between the spatial charge modulation and the magnetic order, which has also been observed in YBCO (e.g. NatComms 7, 11494), Fe5GeTe2 (e.g. PRB104.165101), UTe2 (2207.09491), NdSbTe (2301.13102) and others, but not the direct evidence of BOW. In addition, the data taken at zero magnetic field in figure 6 already show the broken rotational symmetry from the FFT analysis in all energies. One concern is if this could be an extrinsic effect, such as strain from substrates or the tip shape. Then, the in-plane magnetic field may enhance the modulation along one direction. However, it is not possible to tell from the data when the color scale is only marked as high/low and without knowing the direction of in-plane magnetic field in these images.

The symmetry breaking at zero magnetic field is not an extrinsic effect due to, e.g., an asymmetrical tip shape. We have provided additional bias-dependent STM images in Supplementary Fig. 13 (also copied below) taken with a different tip and on a different sample, which also shows the energy-dependent stripe modulation, particularly near the Fermi level.

To quantify the energy dependence of the stripe modulations, we have also provided energy-dependent FFT peak intensity (new Fig. 5g, and Supplementary Fig. S17). The results indicate a complex relationship, along the direction marked by a white dotted line in Fig. 5d, the maximum of the peak intensity shifts from ~ 50 meV below Fermi level to ~ -100 meV upon the application of

an in-plane 2T magnetic field along the direction marked by a red arrow in Fig. 5c topographic image. This shift is also remanent after the removal of the magnetic field. Nevertheless, such a dependence is not observed along the other two directions (Supplementary Fig. S17). These results show that while the stripe modulation is tunable by the in-plane magnetic field, the mechanism is likely complex.

The references suggested are included as Refs. 31-34 The direction of the applied magnetic field is marked in new Fig. 5c by a red arrow.

Overall, these results do demonstrate interesting symmetry breaking phenomena and spatial charge modulation, which appears to be signatures of kagome materials. If the authors remove BOW from the title, tone down their claims and include other possible origins (e.g. CDW or van Hove singularity that have also been observed in other kagome compounds), I will consider to recommend its publication in Nature Communications.

We have changed the title to “Visualizing symmetry-breaking electronic orders in epitaxial

Kagome magnet FeSn films”. We have also revised the manuscript to focus on the trimerization of the Kagome lattice and the stripe modulation tunable by in-plane magnetic fields, and have also included discussions of other possible origins for these observations (pages 7-8), copied below.

“...we discuss electronic orders, including charge and bond orders as the possible origin. For charge density order in non-magnetic Kagome materials, while most studies focus on the impact of the van Hove singularities at the M points^{7,18,20}, recent work highlights the interlayer coupling of the Kagome layers where the interactions between modes at M and L points of the BZ lead to multiple CDWs³⁶, including possibly the nematic order observed in CsV₃Sb₅²¹. Similarly, for magnetic Kagome material FeGe, a recent study suggests that coupling of magnetism and (2 × 2) CDW order can lead to Kekulé-like bond order in the Ge layer³⁷. In the current system of epitaxial FeSn thin film, the coupling between the Sn honeycomb layer and the Fe₃Sn Kagome layer could lead to modifying the Kagome lattice. For example, if the Sn layer exhibits ($\sqrt{3} \times \sqrt{3}$) CDWs, such an order can cause different displacements of the Sn atoms underneath the neighboring triangles of the Kagome lattice, potentially leading to trimerization. Evidence for such enhanced interlayer coupling can be found in the XRD data, where the (002) and (021) peaks of the FeSn are shifted slightly to larger values (Supplementary Figure S2), indicating a smaller c-axis lattice constant. For thinner films, we have reported a strain-induced substantial deformation of the Sn honeycomb on the Sn-termination³⁸ (Supplementary Figures S18-19).”

Reviewer #2 (Remarks to the Author):

The author has performed additional experiments on magnetic field dependence of the density wave order and observed a three-fold symmetry-breaking stripe order phase using scanning tunneling microscopy. The MBE growth result and the scanning tunneling microscopy observed symmetry-breaking density waves are interesting and worth attention. However, the claim on forming bond order waves (BOW) is not convincing. Thus, I can only make recommendations after the questions below are properly addressed.

1. The BOW phase is characterized by alternating strengths of the expectation value of the kinetic-energy operator on the bonds, in other words, hopping parameters [Physical Review B, 65, 155113 (2002)], and the charge density should be uniform [Physical Review B 82, 075125 (2010)]. The requirement of the bond wave order is quite critical and only happens at a tiny region in the phase space at specific fractional filling like 1/3 or 1/4. What is the filling factor here? STM measures mostly charge densities; even if a bond order exists, I am not convinced that STM can tell bond order waves from charge density waves. To my knowledge, the bond order wave has yet to be observed experimentally.

The reviewer is correct that there hasn't been experimental observation of bond order in the Kagome lattice. However, such bond order has been reported for the honeycomb lattice of graphene [Nat. Phys. 12, 950–958 (2016), Nature 605, 51–56 (2022), Science 375, 321–326 (2022)], where it appears as alternating high/low contrasts in STM images and dI/dV maps.

Our observation of the trimerization of the Kagome lattice and energy-dependent contrast reversal between adjacent triangles (Fig. 3) are consistent with bond order. However, the reviewer is correct

that trimerization could arise from other mechanisms. We have now included discussions of possibilities of charge order from the coupling between the Kagome layers and with the underlying Sn layer (pages 7-8), also copied below.

“...we discuss electronic orders, including charge and bond orders as the possible origin. For charge density order in non-magnetic Kagome materials, while most studies focus on the impact of the van Hove singularities at the M points^{7,18,20}, recent work highlights the interlayer coupling of the Kagome layers where the interactions between modes at M and L points of the BZ lead to multiple CDWs³⁶, including possibly the nematic order observed in CsV₃Sb₅²¹. Similarly, for magnetic Kagome material FeGe, a recent study suggests that coupling of magnetism and (2 × 2) CDW order can lead to Kekulé-like bond order in the Ge layer³⁷. In the current system of epitaxial FeSn thin film, the coupling between the Sn honeycomb layer and the Fe₃Sn Kagome layer could lead to modifying the Kagome lattice. For example, if the Sn layer exhibits ($\sqrt{3} \times \sqrt{3}$) CDWs, such an order can cause different displacements of the Sn atoms underneath the neighboring triangles of the Kagome lattice, potentially leading to trimerization. Evidence for such enhanced interlayer coupling can be found in the XRD data, where the (002) and (021) peaks of the FeSn are shifted slightly to larger values (Supplementary Figure S2), indicating a smaller c-axis lattice constant. For thinner films, we have reported a strain-induced substantial deformation of the Sn honeycomb on the Sn-termination³⁸ (Supplementary Figures S18-19).”

2. In Fig. S6f, DFT simulated STM image does not capture the 6-fold symmetry breaking as is shown in the experimental STM image. Can the author comment on it? Can the author provide more detail on the DFT simulation?

The symmetry-breaking orders are likely induced by many-body interactions. It's well known that such interactions cannot be well described at the DFT level, which is an intrinsic drawback of standard DFT calculations. As such, our simulated STM images cannot capture the 6-fold symmetry-breaking characters observed in the experiment.

Our STM simulation using the Tersoff-Hamann approximation (*Phys. Rev. B* **31**, 805 (1985)). The STM tunneling current is proportional to the local density of states of the sample surface at the position of the tip. Therefore, the simulated STM image is the plot of the charge density distribution in a chosen energy window on one horizontal plane above the sample surface. Here, the theoretical energy window is compared to the STM bias voltage, and the vertical position of the horizontal plane is compared to the height of the STM tip. In the simulations, the orbital of the STM tip is considered an isotropic s-wave, and the density is directly obtained from the standard DFT calculations with PBE function using VASP codes. These details are now included in the *Methods* section.

3. Why would the bond order depend on the bias? Why do the strong bond and the weak bond switch from up-triangle-site to down-triangle-site position as a function of energy?

The formation of bond order leads to an insulator. Thus, the filled and empty states are expected to differ for STM imaging and dI/dV mapping. For example, the bond order in graphene also

exhibits a bias-dependent behavior [*Nat. Phys.* **12**, 950–958 (2016), *Nature* **605**, 51–56 (2022), *Science* **375**, 321–326 (2022)]. We have included the following discussion in the revised manuscript:

*“While there haven’t been any reports of bond order in the Kagome lattice, such structure has been observed for the honeycomb lattice of graphene where it appears as alternating high/low contrasts in STM images and dI/dV maps [*Nat Phys.* 12, 950 (2016); *Nature* 605, 51 (2022); *Science* 375, 321 (2022)]. Hence our observation of the trimerization of the Kagome lattice and energy-dependent high/low contrast between adjacent triangles (Fig. 3) can also be explained by the formation of bond order. Such bond order is expected to open a gap, consistent with our observation of the ~ 80 meV gap around the EF (Fig. 1g inset), at which the contrast reversal of neighboring triangles is also observed (cf. Figures 3f vs. 3j).”*

4. I do not understand the assignment of the lattice registry in Fig. 5. As the lattice has 6-fold symmetry and Fig. 5 presents three separate islands, 60 degrees rotation should be equivalent for them. In other words, the relative angle from Fig. 5a and 5d can also be 69.2 degrees. Then rotating Fig. 5e and 5f clockwise by 60 degrees, their trimerization and dimerization direction would be the same as Fig. 5b and 5c, respectively. The same applied to Fig. 5g to 5i. Then the argument in electronic rather than structural structure does not work (line #160 to line #163).

The images in Fig. 5 are taken on different islands, which could have epitaxial relationships that are rotated $\sim 60^\circ$ from each other. The reviewer could be correct that such rotation can lead to the rotation of the bound states and thus does not provide any additional insights into the trimerization of the Kagome lattice. We have now removed the original Fig. 5.

However, the trimerization shows energy-dependent contrast reversal (Fig. 3) and therefore is electronic rather than structural. This is further supported by the magnetic field tunable stripe modulations shown in new Fig. 5.

5. Applying an in-plane magnetic field naturally breaks the 3-fold symmetry into two-fold as the B field can only be applied along one direction. What is the relative angle between the applied B field and the formed stripe orientation? Single q smectic bond order mentioned in Ref. #31 does not rely on the magnetic field. This piece of the experiments is interesting and worth reporting but I do not think it becomes additional evidence for the bond order formation.

We agree with the reviewer that the magnetic field data (new Fig. 5) do not provide additional evidence for bond order. However, it provides strong evidence for the coupling of charge order (that’s responsible for trimerization and stripe formation) with the magnetic field. We have now included discussions of coupling between charge order and magnetism (page 7), also copied below.

“To quantify the evolution of the stripe modulation, the energy-dependent FFT peak intensity is plotted in Fig. 5g and Supplementary Figure S17. Along the direction marked by a white dotted line in Fig. 5d second panel, the maximum peak intensity shifts from ~ 50 meV below Fermi level at zero field to ~ 100 meV at 2 T in-plane field. Interestingly, this shift is also remanent after removing the magnetic field. However, such behavior is not observed in the other two directions (Supplementary Figure S17). These results clearly show a stripe modulation highly tunable by the

magnetic field, indicating the coupling of magnetism with the charge order of the Kagome layer. However, the mechanism is likely complex, similar to other materials such as YBCO³¹, Fe_{5-x}GeTe₂³², UTe₂³³, and NdSb_xTe_{2-x-δ}³⁴, where the coupling between the spatial charge modulation and the magnetic order has also been reported.”

Overall, the claim on bond order wave is not convincing. The author should re-edit the title and the manuscript to focus on reporting the interesting discovery of symmetry-breaking energy and magnetic-field dependent density waves (electronic order) rather than making a big but not convincing claim on bond order waves.

We have changed the title to “Visualizing symmetry-breaking electronic orders in epitaxial Kagome magnet FeSn films”. We have also revised the manuscript to focus on the trimerization of the Kagome lattice and the stripe modulation tunable by in-plane magnetic fields and have also included discussions of other possible origins for these observations in the revised manuscript (pages 7-9).

Reviewers' Comments:

Reviewer #1:

Remarks to the Author:

The authors have addressed my previous concerns and revised the manuscript accordingly. I can now recommend its publication.

Minor point: The authors should mark the direction of the in-plane magnetic field in Fig S14 and S15 in the final version.

Reviewer #3:

Remarks to the Author:

In Fig. 5g and Fig.S17, the author plotted the FFT peak intensity as a function of energy. However, one needs to be careful when comparing the absolute FFT intensity across energies. This is because these maps are taken independently at different biases (i.e., different tip-to-sample tunneling conditions), and FFT peak intensity can be sensitive to tip-to-sample tunneling conditions. Typically, one can normalize them to the Bragg peak intensity. However, here, the Bragg peak intensity also carries the information of the electron order and does not remain a constant value. Given this consideration, I suggest that the author normalize the FFT peak intensity to their corresponding averaged background intensity before comparing them across energies.

Other than that, the author has addressed my question, and I can recommend its publication in Nature Communication.

REVIEWERS' COMMENTS

Reviewer #1 (Remarks to the Author):

The authors have addressed my previous concerns and revised the manuscript accordingly. I can now recommend its publication.

Minor point: The authors should mark the direction of the in-plane magnetic field in Fig S14 and S15 in the final version.

We thank the reviewer for the suggestion and have marked the direction of the in-plane magnetic field in updated Figs. S14, S15 and S16.

Reviewer #3 (Remarks to the Author):

In Fig. 5g and Fig.S17, the author plotted the FFT peak intensity as a function of energy. However, one needs to be careful when comparing the absolute FFT intensity across energies. This is because these maps are taken independently at different biases (i.e., different tip-to-sample tunneling conditions), and FFT peak intensity can be sensitive to tip-to-sample tunneling conditions. Typically, one can normalize them to the Bragg peak intensity. However, here, the Bragg peak intensity also carries the information of the electron order and does not remain a constant value. Given this consideration, I suggest that the author normalize the FFT peak intensity to their corresponding averaged background intensity before comparing them across energies.

Other than that, the author has addressed my question, and I can recommend its publication in Nature Communication.

We thank the reviewer for the suggestion. In Fig. 5g and Fig. S17, the dI/dV maps are collected at the same tip-sample distance. We have normalized the FFT peak intensity to their corresponding averaged background intensity. We have updated Figs. 5g and S17 in the revised manuscript.